# Quantification of Airborne Particulate Matter and Trace Element Deposition on *Hedera helix* and *Senecio cineraria* Leaves

**DOI:** 10.3390/plants13172519

**Published:** 2024-09-07

**Authors:** Anabel Saran, Mariano Javier Mendez, Diego Gabriel Much, Valeria Imperato, Sofie Thijs, Jaco Vangronsveld, Luciano Jose Merini

**Affiliations:** 1Consejo Nacional de Investigaciones Científicas y Técnicas (CONICET), Agencia de Investigación Cientifica, Santa Rosa PC6300, Argentina; diego.much@live.com.ar (D.G.M.); lucianomerini@yahoo.com.ar (L.J.M.); 2Consejo Nacional de Investigaciones Científicas y Técnicas (CONICET), Facultad de Agronomía, Universidad Nacional de La Pampa, Santa Rosa PC6300, Argentina; marianomendezz@hotmail.com; 3Environmental Biology, Centre for Environmental Sciences, Hasselt University, 3590 Diepenbeek, Belgium; vale.imperato@gmail.com (V.I.); sofie.thijs@uhasselt.be (S.T.); jaco.vangronsveld@uhasselt.be (J.V.); 4Department of Plant Physiology and Biophysics, Institute of Biological Sciences, Maria Curie Sklodowska University, 20-031 Lublin, Poland

**Keywords:** particulate matter, air pollution, trace elements, bio-monitors, plants

## Abstract

In both developed and developing countries, atmospheric pollution with particulate matter (PM) remains an important issue. Despite the health effects of poor air quality, studies on air pollution are often limited by the high costs of continuous monitoring and the need for extensive sampling. Furthermore, these particles are often enriched with potentially toxic trace elements and organic pollutants. This study evaluates both the composition of atmospheric dust accumulated during a certain timespan on *Hedera helix* and *Senecio cineraria* leaves and the potential for their use as bio-monitors. The test plants were positioned near automatic air quality monitoring stations at four different sites with respectively high, moderate and low traffic intensity. The gravimetric deposition of PM10 and PM2.5 on leaves was compared with data recorded by the monitoring stations and related to the weather conditions reported by Argentina’s National Meteorological Service. To determine the presence of trace elements enriching the PM deposited on leaves, two analytical techniques were applied: XRF (not destructive) and ICP (destructive). The results indicated that only in the unpaved street location (site 2) did PM10 and PM2.5 concentrations (90 µg m^−3^ and 9 µg m^−3^) in the air exceed more than five times WHO guidelines (15 µg m^−3^ and 5 µg m^−3^). However, several trace elements were found to be enriching PM deposited on leaves from all sites. Predominantly, increased concentrations of Cd, Cu, Ti, Mn, Zn and Fe were found, which were associated with construction, traffic and unpaved street sources. Furthermore, based on its capability to sequester above 2800 µg cm^−2^ of PM10, 2450 µg cm^−2^ of PM2.5 and trace elements, *Senecio cineraria* can be taken into consideration for adoption as a bio-monitor or even for PM mitigation.

## 1. Introduction

According to a report by the European Environment Agency (EEA, 2017), air pollution is a major environmental and social problem which has led to numerous challenges in terms of management and mitigation. Effective action to reduce the impact of air pollution requires a good understanding of how pollutants are transported and transformed in the atmosphere, and how they affect humans, ecosystems, the climate and, subsequently, society and the economy [1]. Suspended particulate matter (PM) is a significant atmospheric pollutant with severe public health effects, particularly in urban areas which are heavily affected by emissions from vehicles, industry and other sources of air pollution [2]. The correlation between emissions and pollution levels differs depending on the city or even the part of a city, whereby infrastructure and urban planning determine the emission pattern while meteorology and topography determine dispersion and transformation [3].

Inhalable particulate matter (PM) refers to particles with a diameter of less than 10 μm. These particles have a complex composition, including organic constituents, inorganic salts, and trace elements [4]. Different studies have associated PM10 with severe respiratory diseases, such as asthma [5], lung cancer [6] and chronic obstructive pulmonary disease [7], as well as with cardiovascular diseases such as stroke, deep vein thrombosis, coronary events, myocardial infractions and atherosclerosis [8]. Between 1990 and 2019, worldwide deaths due to air pollution increased by 2.62%. Furthermore, over 90% of the global population resides in areas that have not reached the air quality standards established by the WHO (2021) [2].

Despite the impact of air quality on health, studies of air pollution are often restricted by the high costs of monitoring instruments and the limited scale of sampling. Drawbacks to the use of conventional monitoring equipment in the field are the high price, large size, heavy weight and energy consumption [9]. For these reasons, bio-monitoring, using plants or lichens to estimate air quality, has been proposed as a cost-effective and environmentally friendly approach which can be an alternative to physical and chemical analytical methods of air pollution monitoring [10]. In this way, plants were introduced as bio-monitors of trace element accumulation due to their efficiency in trapping particulate matter [11]. Air pollutants can be bounded in and on the cuticula and, eventually, taken up by plants via stomata, or indirectly by uptake via the roots after deposition of the air pollutants on the soil [12]. After penetration, the particles clog the stomata and decrease the foliar pH due to the presence of sulphate (SO_4_^2−^) and nitrate (NO_3_^−^) ions in the dust [13]. Atmospheric dust deposition on leaves is mainly influenced by the plant species (evergreen or deciduous; composition and thickness of the wax layer) and the specific structure of their leaves (e.g., leaf size, shape, roughness and the presence of trichomes), as well as the meteorological conditions (air humidity, rainfall and wind) and source-specific particle features (e.g., particle size distribution) [14]. Several plants have been used as bio-monitors of air pollution: some of them due to their resistance, e.g., *Pleurozium schreberi*, *Sphagnum fallax* and *Dicranum polysetum* [15], and others due to their sensitivity, e.g., *Nicotiana tabacum*, *Petunia hibrida*, *Ricinus comunis* and *Trifolium pretense* [11]. In this context, the improvement of green spaces or shelter belts by planting suitable and tolerant species selected for the specific area can catch air pollutants and mitigate the pollution levels.

This study aimed to assess the levels of air pollution in sites with high, moderate and low traffic intensity, relating bio-monitors to automatic monitoring data, as well as to assess the usefulness of destructive and non-destructive analytical techniques for the quantification of trace elements using outdoor exposed *Hedera helix* and *Senecio cineraria* plants.

## 2. Results

### 2.1. Daily Meteorological Conditions and Air Quality Monitoring Stations

The highest temperature (higher than 30 °C) was recorded in the summer season in January (13 January 2022) (Figure 1). Relatively strong winds appeared between 25 October 2021 and 10 November 2021, being 20 times stronger than the average for the rest of the recorded days. Humidity fluctuated between 31.75% and 96.46%, the latter measurement being consistent with the peak precipitation recorded (62.4 mL/m^2^; 3 December 2021). 

The monthly mean PM concentrations recorded by automatic monitors are presented in Figure 2. Site 2 (moderate car traffic and unpaved streets) is the only site exceeding the PM2.5 threshold recommended by the WHO (2021). September and October (spring season) were the months with the highest PM2.5 concentrations recorded. Also, the monthly mean PM10 concentrations (Figure 2) recorded at Site 2 exceeded the WHO threshold, as did those at Site 3 (moderate car traffic); however, the mean concentrations recorded at this site were significantly lower than those recorded at Site 2, showing that the biggest contribution of PM10 mainly originated from the unpaved streets. In Appendix A, daily average concentrations recorded for PM10 and PM2.5 at each site are presented. 

A Spearman correlation matrix (Figure 3) showed a positive correlation (*p* < 0.05) between temperature and PM10 and PM2.5 for sites 3 and 4. A negative correlation (*p* < 0.05) was observed between humidity and PM10 across all sites, while a slight negative correlation (*p* < 0.05) between humidity and PM2.5 was seen for sites 2 and 3. Only at site 4 was a positive correlation (*p* < 0.05) (Figure 3) identified for PM10 and wind speed. Concerning precipitation, a negative correlation (*p* < 0.05) was observed across all sites for PM10, whereas this correlation was evident only at sites 2 and 3 when considering PM2.5.

### 2.2. Gravimetric Quantification of Particulate Matter Deposited on Leaves

Table 1 shows the mean concentrations, and standard errors, of PM10 and PM2.5 deposited on *Hedera helix* and *Senecio cineraria* leaves. ANOVA indicates significant differences in the amounts of PM10 quantified from the leaf surfaces of *Hedera helix* and *Senecio cineraria* (*p* > 0.05). *Senecio cineraria* sequestered between two and eight times more PM10 per cm^2^ than *Hedera helix*. This can be explained by the micromorphology of *Senecio cineraria* leaves (Appendix A), which appeared more rugged than *Hedera* leaves. We also found significant differences in PM2.5 deposited at site 3 (*p* > 0.05), where both species sequestrated the highest amounts of PM2.5 (*Hedera*: 965 µg·cm^−2^; *Senecio*: 2450 µg·cm^−2^). For both species, significant increases in PM10 and PM2.5 were observed after 3 months at site 3 (*p* > 0.05) and after 6 months at site 4 (*p* > 0.05). At sites 1 and 2, species behaved differently. At site 1, *Hedera* did not show any significant accumulation of PM10 and PM2.5; in contrast, for *Senecio* the amount of PM10 increased significantly (*p* > 0.05) after 3 months while the amount of PM2.5 decreased. After 6 months at site 2, significant increases were observed for PM2.5 on *Hedera* and for PM10 on *Senecio*.

### 2.3. Leaf Surface Elemental Composition: XRF and ICP

The means (n = 5) of the XRF spectra of *Hedera helix* and *Senecio cineraria* are presented in Figure 4 and Figure 5 respectively. Appendix A presents the mean element distributions and deviations obtained. K and L X-ray emission lines were preset to perform the quantification. Initially, an assessment was made of the elements present on the nonexposed clean leaves (0 months) (gray spectra in Figure 4 and Figure 5). The elements Al, Si, P, S, K, Ca, Mn and Fe were present in all non-exposed clean leaves for both *Hedera helix* and *Senecio cineraria* plants, of which Si, K and Ca were the most abundant elements. After 3 and 6 months, the predominant elements determined by XRF included Ti, Zn and Fe. The Ti mean element distribution was similar for *Hedera* and *Senecio* located at site 2. Also, *Senecio* plants located at site 1 showed Ti accumulation. Zn accumulation was observed in *Hedera* plants located at site 2 and *Cineraria* plants located at site 1. Fe found in *Senecio* leaves (normalized weight %) was more than three times higher than that found on *Hedera*. However, the observed variabilities between leaves from the same species and exposure time were larger than expected (Appendix A).

Table 2 shows the mean element concentrations, and standard deviations, of the elements quantified by ICP in *Hedera helix* and *Senecio cineraria* leaves. Moreover, Figure 6 shows the biplot based on the first two principal components, which explain around 99.42% of the variability of the PCA analysis. Sample distributions were categorized by (A) site and plant species, (B) exposure time in months, while (C) the loading plot emphasizes elements that influence sample distributions. The same elements measured by XRF were detected by ICP, except for Ti, Si and Al. Cd was detected by ICP analyses in *Hedera helix* and *Senecio cineraria* leaves located at sites 1, 2 and 3. The highest accumulation was recorded after 3 months at site 2. Cd was not detected by XRF. The high K peak in the XRF spectra may likely overlap with Cd lines, obstructing its detection and quantification in case Cd is present in low concentrations. Also, Cu was only detected by ICP. 

*Senecio cineraria* samples from sites 1 and site 2 were separated from the rest of the samples (Figure 6A). Site 2 shows a dispersion across PC1 explaining 96.28% of the total variance and site 1 demonstrates a dispersion across PC2 explaining 3.14% of the total variance. Figure 6C suggests a direct influence of Zn in the separation of *Senecio cineraria* located in site 1 across PC2 and the influence of Fe in the separation of *Senecio cineraria* located in site 2 across PC1. This indicates that PM-Zn is more typical of high traffic zones (Site 1) and PM-Fe is abundant in unpaved street locations (Site 2). The loading values of Zn and Fe in the two first principal components of PCA explain more than 99.42% of the data variation, suggesting that these two elements were the main factors causing the difference in element composition between samples. Furthermore, Cu and Mn displayed an influence in the coordinate origin area, indicating that samples from sites 3 and 4 can be enriched with these elements. Figure 6B shows that samples taken at time points 0 and 3 months have a similar distribution, suggesting a variation in trace element concentrations after the third month. After 3 months of exposure, leaf trace element concentrations exhibited significant decreases (*p* > 0.05) for Cd, Ca and Na (Table 2) and increases for Zn, Cu and Fe.

Welch’s two sample t-test comparison of a total of nine (common) elements (Ca, Fe, K, Mg, Mn, Na, P, S and Zn) measured by XRF and ICP showed that the average leaf surface elemental concentrations of Ca, Fe, Na and Zn were statistically similar (*p* > 0.05) when determined by both techniques for both plant species. P, Mg and S average concentrations were higher when determined by ICP, while the average concentrations for K and Mn were always higher after using XRF (Figure 7).

## 3. Discussion

Meteorological conditions were recorded by Argentina’s National Meteorological Service, Creative Commons 2.5 Argentina License (Figure 1). According to the Thornthwaite climate classification, the climate of the center and south of this agricultural zone (Santa Rosa) is dry subhumid, with little or no excess water, cold temperate mesothermal with a summer concentration of thermal efficiency less than 48% [16]. Temperature, precipitation, humidity and wind speed were related to concentrations recorded by automatic monitors (Figure 2) by means of a Spearman correlation test. It has been reported that a lower temperature leads to higher humidity, which leads to an increase of both PM2.5 and PM10 deposition and absorptions by plants [17]. The effect of air humidity on deposition is due to the fact that particles are mainly hygroscopic, and their size varies as a result of the absorption or discharge of water. In return, this leads to a change in their deposition properties as a function of diameter [18]. Our results showed a positive correlation between temperature and PM10 and PM2.5 for sites 3 and 4 and a negative correlation between humidity and PM10 across all sites (Figure 3). Regarding wind speed, Mei et al. [19] reported that an increase in wind speed enhanced particle transport and reduced local particle concentrations; however, it did not affect the relative location of high particle concentration zones, which are more related to building height and design. In accordance with Mei et al., we found a positive correlation between wind speed and PM10 (Figure 3) only at site 4 (a rural area). Additionally, Liu et al. [20] reported that precipitation has a certain wet scavenging effect on PM2.5 and PM10, and the scavenging effect on PM10 is higher than that on PM2.5. They concluded that the scavenging effect of precipitation on PM10 is closely related to the initial concentration of PM10 before precipitation. The higher the initial concentration of PM10, the greater the removal by precipitation. A negative correlation was observed across all sites between precipitation and PM10, regardless of the initial concentration of PM10, whereas this correlation was evident only at sites 2 and 3 when considering PM2.5 (Figure 3). 

The gravimetric quantification of PM deposited on leaves (Table 1) indicates a significant difference in the amount of PM10 sequestered by *Senecio cineraria* compared to *Hedera helix*. Deposition was already reported to be mainly affected by the shape of the plant and the structure of the leaves or needles [21]. Also, *Senecio cineraria* leaves are covered with fine matted hairs, giving them a felted or woolly appearance [22]. The accumulation of atmospheric PM reported by Castanheiro et al. [23] was shown to be species-specific (hedera accumulated more than strawberry) rather than influenced by the buildup of atmospheric dust. Urban, rural and industrial areas were studied using *Celtis occidentalis* and *Trientalis europaea* leaves in the city of Debrecen (Hungary) [11]. In contrast to our findings, an increasing amount of fine dust deposited on leaves was found in the urban area compared to the industrial and rural sites, suggesting that the higher vehicular traffic had a notable effect on dust emission and deposition on leaves at the urban site. Similarly, in the city of Gandhinagar, India, Chaudhary et al. [24] found higher dust depositions on tree leaves in zones with intense traffic compared to commercial and residential zones. However, we did not find additional PM deposition at the urban site (site 1) compared to the peri-urban sites (sites 2 and 3) and the rural site (site 4). This might be due to the fact that the diffusion of PM was not hindered by tall rows of buildings at the urban site. There was no correlation between the location where plants sequestered the highest amount of PM (site 3, Table 1) and the location where the monitors reported the highest amount of PM (site 2, Figure 1). Outcomes obtained with both techniques (automatic monitors and bio-monitors) are not associated; however, they can be used as complementary tools to elucidate the complex, multifactorial process of PM diffusion and deposition. The major advantage of automatic monitors is that meteorological factors do not affect the accuracy of the obtained results; however, it is not possible to measure trace element enrichment. This is possible using *Senecio cineraria* leaves. Świsłowski et al. [15] compared biological monitoring using mosses and air filter standard sampling. Their research indicates that the results obtained by the two methods (active biomonitoring and dust deposited on the filter) have different applications. Mosses accumulate bioavailable forms of metals and are affected by many external factors during exposure (thus changing their degree of vitality); therefore, the results were different from those obtained with an automatic device. On the other hand, after characterizing the heavy metal concentrations in a total of 540 samples from four ecosystem compartments (plant leaves, foliar dust, surface soil and subsoil), Li et al. [25] concluded that foliar dust reflected pollution of atmospheric particulate matter most reliably among the four ecosystem compartments that were investigated. 

After 3 and 6 months of exposure, the predominant elements found to be enriching PM deposited on *Hedera* and *Cineraria* leaves included Ti, Zn and Fe, using XRF (Figure 4 and Figure 5). Some researchers have reported that Fe could be associated with soil resuspension since this element is a typical soil constituent [26]. Zn and Fe can be derived from exhaust and non-exhaust road traffic [27]. Zn is also associated with tire tread dust [28], while Ti is widely used in paint as a UV filter, as well as in plastic [29], and has been detected before in outdoor air [30,31]. Similar results were reported by Castanheiro et al. [23], who studied the accumulation of atmospheric dust on ivy and strawberry leaves using XRF and concluded that XRF offers many advantages for multi-element, non-destructive analysis, which can be performed directly on the sample, at a relatively low cost and with rapid output. However, disadvantages are the heterogeneity of plant material and matrix effects [23]. Particularly when samples do not meet the condition of thin-film, self-absorption effects arise that complicate the process of matrix calibration required for quantitative analysis [32]. In our study, these effects were also observed as the variabilities between leaves from the same species, and exposure times were larger than expected (Appendix A). Santos et al. [26] used *Nerium oleander* L. leaves as a bio-monitor to evaluate levels of environmental pollutants in a sub-region in the Metropolitan Area of Rio de Janeiro City (Brazil) through XRF. In their study, they highlighted the association between Fe, Cu, Zn and Pb and vehicle and industrial emission sources and the usability of the XRF technique for environmental pollution analysis. Furthermore, according to Hulskotte et al. [33], vehicle braking systems are one of the most significant sources of Cu particles. In our study, the enrichment of Cd, Cu, Fe, Mn and Zn in leaves was found to result from the accumulation of atmospheric dust by means of ICP (Table 2). 

For the interpretation of field data and evaluation of trace element air pollution, reference values are needed. “Reference Plant” was proposed by Markert [34] and describes the average content of all the inorganic elements found in plants. The following values could be considered ‘normal’ metal concentrations in leaves of plants from uncontaminated environments: 0.05 mg·kg^−1^ Cd, 10 mg·kg^−1^ Cu, 150 mg·kg^−1^ Fe, 200 mg·kg^−1^ Mn and 50 mg·kg^−1^ Zn. These values of element concentration can be further used to establish the reference point of the “chemical fingerprint” [35]. In our study, Cd, Fe and Zn exceeded these thresholds (Table 2). These elements may become a threat to human health and/or the environment. Chronic exposure to low levels of heavy metals can cause serious health effects in the long term [36]. Gehring et al. [37] described the adverse effects of PM constituents, in particular Si, K, Fe, Cu and Zn, on asthma, rhinitis, allergic sensitization, and lung function in schoolchildren.

Finally, results obtained by XRF (not destructive) and ICP (destructive) analytical techniques were compared by means of Welch’s two sample t-test (Figure 7). Castanheiro et al. [23] also compared the results obtained by XRF and HRICP-MS techniques from ivy and strawberry leaves exposed to atmospheric dust. For a total of ten (common) elements (Si, K, Ca, Ti, Cr, Fe, Cu, Rb, Sr and Pb), they observed that concentrations were always higher when samples were investigated using XRF in comparison with ICP-MS. However, those ten elements were detected for most analysed leaves with ICP-MS, which was not the case for XRF. Therefore, they consider the accumulation of elements to be most accurate after quantification by HR-ICP-MS. In contrast, our study showed that XRF and ICP results were statistically similar for Ca, Fe, Na and Zn. While Cd and Cu were detected by ICP only due to K peaks overlapping for XRF and Ti, Si and Al were detected by XRF only due to the heterogeneity of plant material and low detection limits for ICP. 

Although it is necessary to perform more research to elucidate whether these plant species can be used as bio-monitors in areas with higher pollution levels, *Cineraria*’s features make it a promising candidate to be adopted as a bio-monitor or even for PM mitigation when planted as a green belt.

## 4. Materials and Methods

### 4.1. Selection of Plant Species

*Hedera helix* is an evergreen climbing plant with alternating leaves, 50–100 mm long, with 15–20 mm long petioles. It possesses palmately five-lobed juvenile leaves on creeping, climbing stems and unlobed cordate adult leaves on fertile flowering stems exposed to full sun (Metcalf, 2005). This species was chosen for its known capacity to capture a wide spectrum of PM fractions (0.2–2.5 μm; 2.5–10 μm; 10–100 μm) [38,39].

*Senecio cineraria* is a white-woolly, heat- and drought-tolerant evergreen subshrub. The leaves are pinnate or pinnatifid, 5–15 cm long and 3–7 cm broad, stiff, with oblong and obtuse segments, and like the stems, thinly to thickly covered with long grey-white to white hairs. The tomentum is thickest on the underside of the leaves, and can wear off on the upper side, leaving the top surface glabrous with age [40]. This species was chosen as its capacity to accumulate PM has not yet been evaluated and the characteristics of its leaves make it a good candidate to sequester PM.

### 4.2. Sites Selected, Air Quality Monitoring Stations and Daily Meteorological Conditions

*Hedera helix* and *Senecio cineraria* plants were obtained from a plant nursery (Agropecuaria, Santa Rosa, Argentina). After the collection of non-exposed cleaned leaves, five plants of each species were placed at the final locations. Four locations in Santa Rosa, La Pampa, Argentina, with different anthropogenic impacts were selected for the study. Site 1: University, an urban area with intense car traffic based on Google Maps Traffic statistics (−36.625325 Lat., −64.293103 Long.); Site 2: Calo Street, a suburban area with moderate car traffic and unpaved streets (−36.647888 Lat., −64.276939 Long.); Site 3: Felice Street, located in a residential area with moderate car traffic (−36.627074 Lat., −64.323317 Long.); and Site 4: Agronomic Campus, located 5 km from Santa Rosa city, a rural area, with little car traffic (−36.548932 Lat., −64.299345 Long.) (Figure 8). Plants were placed next to an automatic air quality measuring station equipped with a laser scattering Sensor SDS-011 (Nova Fitness, Shandong, China) (Appendix A). Daily meteorological conditions (temperature, precipitation, humidity and wind speed) were recorded by Argentina´s National Meteorological Service, Creative Commons 2.5 Argentina License. Plants were watered once a week to prevent drought stress. The watering was performed avoiding any physical contact with the leaves. Leaf sampling was conducted every three months during a period of six months, in spring (15 December 2021) and in summer (16 March 2022). The leaves were sampled at 10 cm above the soil, to avoid direct soil contamination and to standardize any potential influence from resuspension of the soil in the pots. Three leaves per plant were collected per sampling location (n = 60). 

### 4.3. Gravimetric Quantification of Particulate Matter Deposited on Leaves

An Erlenmeyer was filled with 50 mL of ultrapure water and one leaf was added. The leaf was stirred in an orbital shaker for 60 min at 270 rpm as described previously [41]. Subsequently, PM fractions were separated using Type 91 Whatman ashless filters with 10 µm retention and Type 42 with 2.5 µm retention. Before use, the filters were dried overnight in an oven at 60 °C, and their weight was determined to correct for air humidity. After filtration, filters were dried and post-weighed using a PIONER precision balance (OHAUS, Lindavista, Mexico) to calculate the weight of PM in each fraction of every sample. The leaf surface area was determined using Image J Analysis System [42], which allowed for the expression of the amount of PM as mg cm^−2^ leaf area.

### 4.4. Leaf Surface Elemental Composition: XRF and ICP

Leaf samples were analyzed for their elemental composition using X-ray Fluorescence (XRF) and Inductively Coupled Plasma-Atomic Emission Spectrometry (ICP). First, leaf samples were analyzed by XRF for the element range Na–Ba. An XRF benchtop spectrometer (M4 Tornado, Bruker^®^, Germany) equipped with an Rh X-ray tube with polycapillary optics and an XFlash^®^ detector providing an energy resolution of better than 145 eV was employed. For the analyses of (i) Mg–Fe we used a tube voltage of 10 kV, a current of 300 mA, and a live time of 500 s; and for (ii) Ti–Ba: 40 kV, 50 mA, 1000 s. The measured XRF spectra in each pixel of the XRF maps were deconvoluted using software supplied with the M4 Tornado (version 1.5.2.45). Spectrum energy calibration was performed daily, before the analysis of each batch, by using a copper (Cu) and zirconium (Zr) Bruker^®^ calibration standard block. Analytical quality control was ensured by means of the analysis of steel certified reference material (SN0163). Data were normalized against a conservative element and the results were given in % weight normalized to 100%.

Secondly, the same leaves were digested with 70% HNO_3_ in a heat block and dissolved in 5 mL of 2% HCl using the USEPA 3050B Acid Digestion of Sediments, Sludges, and Soils (Environmental Protection Agency [EPA] 1996a). The concentrations of elements were determined by means of inductively coupled plasma-atomic emission spectrometry (HR-ICP-OES, Agilent Technologies, 700 series, Brussels, Belgium). Blanks (only HNO_3_) were included. Also, certified reference materials, BCR cabbage and 1570a spinach leaves, were included in each batch. The recoveries obtained were in the range of 40–67% for Cd, 77–90% for Cu, 79–90% for Mn, 61–70% for Pb and 79–91% for Zn.

### 4.5. Statistical Analysis

Data normal distribution was verified with the Shapiro–Wilk test and the homogeneity of variances was confirmed by a Levene test. The differences between samples were tested using analysis of variance (ANOVA) for each variable. When group variances were unequal, the Games–Howell method was used for pairwise comparison between the groups. A two-sample t-test was applied to analyze months and species categories. Statistical analysis to compare possible significant differences between XRF and ICP techniques was carried out by means of Welch’s *t*-test. Principal component analysis (PCA) was carried out to identify the potential contributions of Cd, Cu, Fe, Mn, and Zn and exposure time (0, 3 and 6 months) at each site. Additionally, Spearman correlation analysis was performed to assess correlations between meteorological variables (temperature, wind speed, humidity and precipitation) and PM fractions (10 and 2.5) at each site. PCA was displayed by Matlab (version R2022a) (The MathWorks Inc., Natick, MA, USA), while the correlation matrix was generated using R (version 4.2.2).

## 5. Conclusions

Outcomes obtained with automatic monitors and bio-monitors are not associated; however they can be used as complementary tools to elucidate the complex, multifactorial process of PM diffusion and deposition. The accumulation of atmospheric PM was shown to be species-specific rather than influenced by the traffic intensity and the atmospheric dust levels. *Senecio cineraria* sequestered between two and eight times more PM10 per cm^2^ than *Hedera helix*. Humidity was the climatic variable that strongly negatively influenced PM recorded by monitors. The unpaved street at site 2 was the only location where PM10 and PM2.5 exceeded WHO guidelines. At the high-traffic site 1, Zn enriched the PM sequestered by plant leaves and Fe was observed on plant leaves located close to unpaved streets. Furthermore, using XRF, Ti was identified on plant leaves, probably originating from construction activities. XRF and ICP results were statistically similar for Ca, Fe, Na and Zn. However, Cd and Cu were detected by ICP only and Ti, Si and Al were detected by XRF only. Therefore, the two techniques have complementary weaknesses. Based on our results, *Senecio cineraria* may be considered a bio-monitor or even part of a PM mitigation strategy.

## Figures and Tables

**Figure 1 plants-13-02519-f001:**
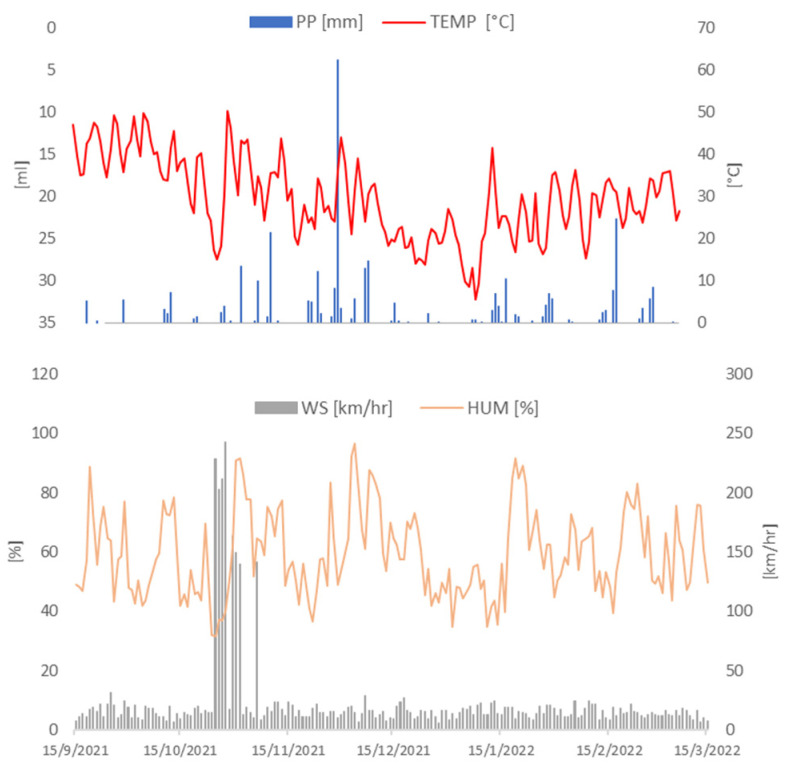
Measurements from an automatic weather station of the National Weather Service of Argentina at Santa Rosa Aero station. Humidity (orange line), wind speed (grey bars), temperature (red line) and precipitation (blue bars) recorded between 15 September 2021 and 15 March 2022.

**Figure 2 plants-13-02519-f002:**
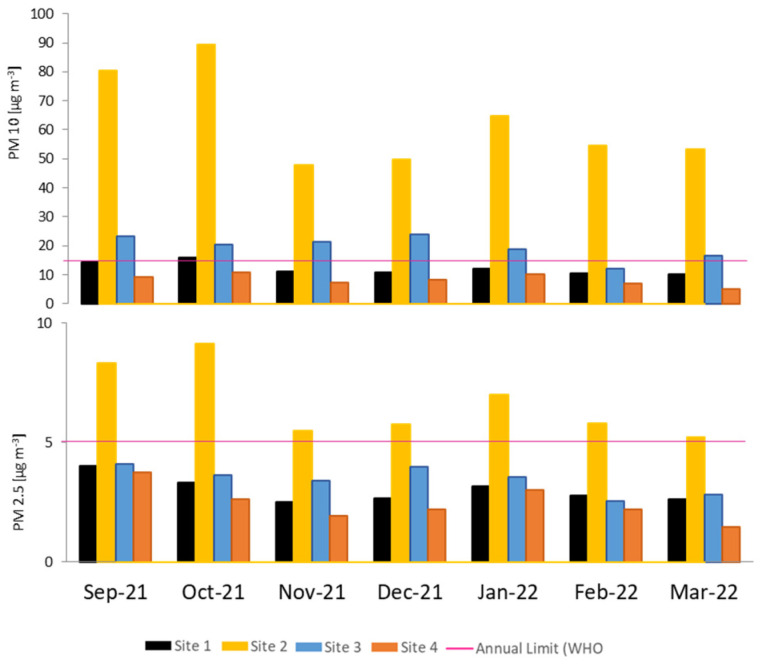
Average concentrations of PM 10 and PM 2.5 recorded monthly at each site (n = 5). The pink line represents the WHO recommended annual limit (PM10 = 15 µg m^−3^; PM2.5 = 5 µg m^−3^).

**Figure 3 plants-13-02519-f003:**
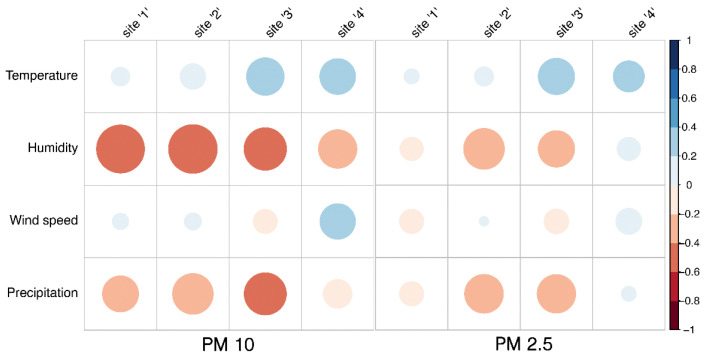
Spearman correlation matrix, pairwise relationships between meteorological variables and PM concentrations recorded by monitors located at the four sites. Circle sizes dynamically adjust based on the magnitude of correlation, and the color gradient indicates the strength and direction of correlations, from negative (red) to positive (blue).

**Figure 4 plants-13-02519-f004:**
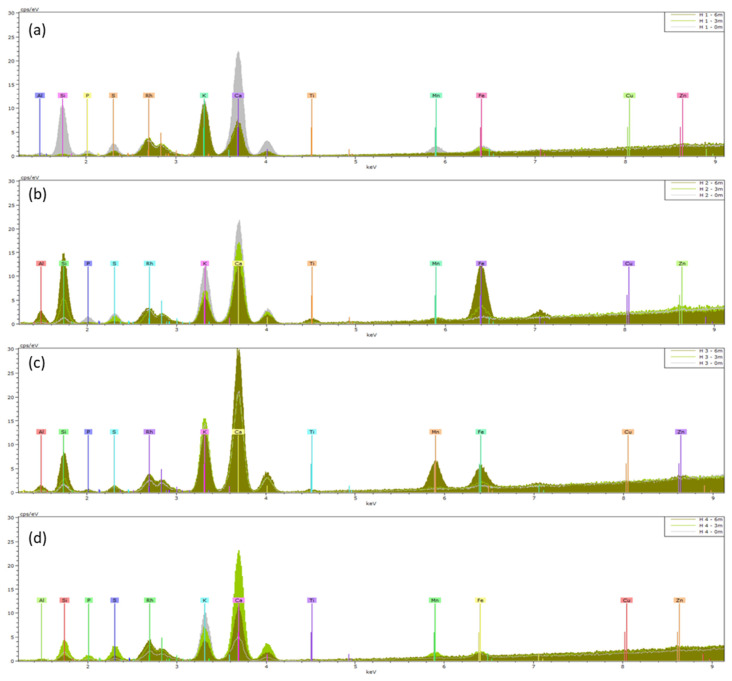
Means (n = 5) of XRF spectra of *Hedera helix* (H) leaves collected from (**a**) site 1, (**b**) site 2, (**c**) site 3 and (**d**) site 4. Leaves were analyzed before (0 m) and after 3 and 6 months of exposure (3 m and 6 m). The KeV of the peaks shows which elements are present, and the height of a peak indicates the abundance of that element.

**Figure 5 plants-13-02519-f005:**
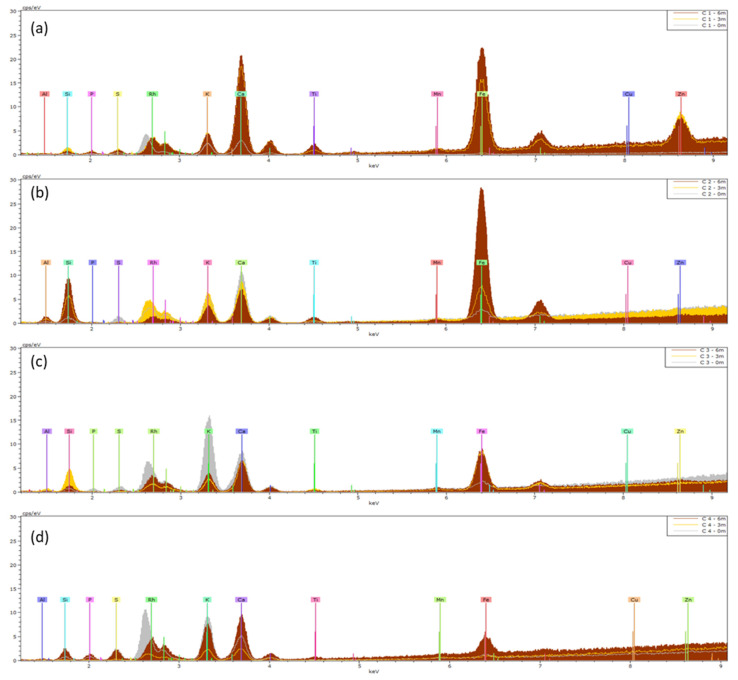
Means (n = 5) of XRF spectra of *Senecio cineraria* (C) leaves originating from (**a**) site 1, (**b**) site 2, (**c**) site 3 and (**d**) site 4. Leaves were analyzed before (0 m) and after 3 and 6 months of exposure (3 m and 6 m). The KeV of the peaks shows which elements are present, and the height of a peak indicates the abundance of that element.

**Figure 6 plants-13-02519-f006:**
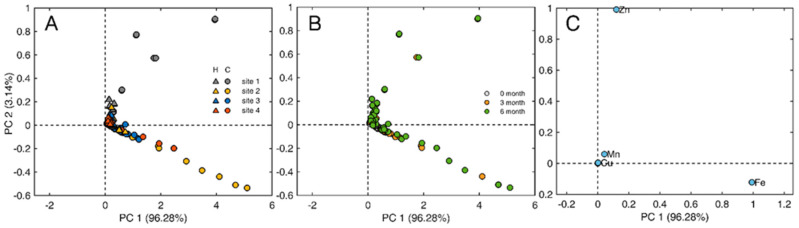
Score plot of the first two PCs obtained by PCA illustrating sample distributions based on (**A**) site and plant species (‘C’ for *Senecio cineraria* and ‘H’ for *Hedera helix*) and (**B**) exposure time in months. (**C**) Loading plot highlighting elements with the main influence on the sample distribution.

**Figure 7 plants-13-02519-f007:**
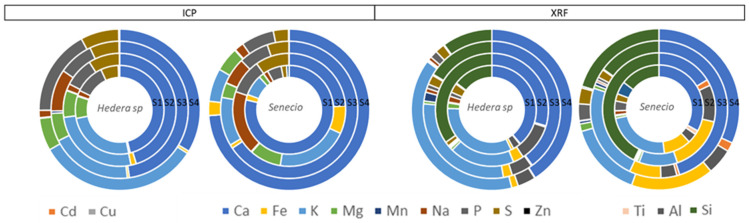
Pie charts of average leaf surface elemental concentration measured by ICP and XRF for *Hedera* helix and *Senecio cineraria* plants after 6 months of exposure at sites 1, 2, 3 and 4. Cd and Cu (left side) were only detected by ICP. Al and Si (right side) were only detected using XRF.

**Figure 8 plants-13-02519-f008:**
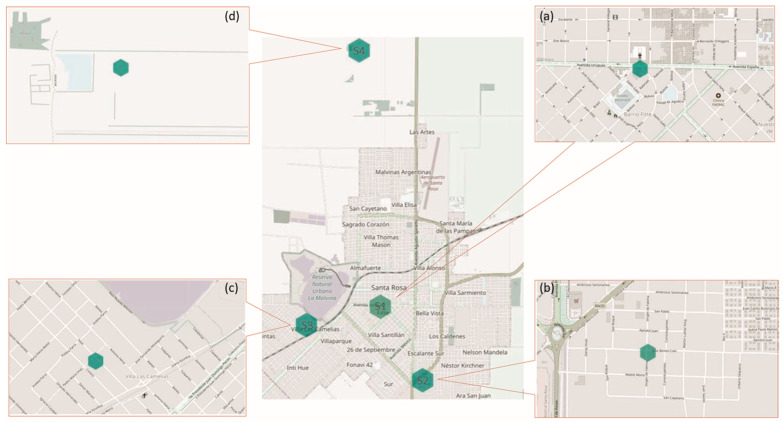
Site locations, Santa Rosa City, La Pampa province, Argentina. (**a**) Site 1, an urban area with high intensity of car traffic; (**b**) Site 2, a suburban area with moderate car traffic and unpaved streets; (**c**) Site 3, a residential area with moderate car traffic; and (**d**) Site 4, a rural area with low intensity of car traffic, based on Google Maps Traffic.

**Table 1 plants-13-02519-t001:** PM10 and PM2.5 (µg·cm^−2^) sequestered on the leaf surface.

Plant	Time	Site 1		Site 2		Site 3		Site 4	
		PM10	PM2.5	PM10	PM2.5	PM10	PM2.5	PM10	PM2.5
*Hedera helix*	0 m	485 ± 96 ^a^	681 ± 125 ^a^	930 ± 175 ^a^	115 ± 44 ^a^	60 ± 26 ^a^	30 ± 23 ^a^	151 ± 51 ^a^	109 ± 90 ^a^
	3 m	341 ± 83 ^a^	552 ± 588 ^a^	699 ± 274 ^ab^	670 ± 577 ^ab^	946 ± 312 ^b^	965 ± 343 ^b^	175 ± 81 ^a^	215 ± 90 ^a^
	6 m	412 ± 43 ^a^	505 ± 465 ^a^	501 ± 330 ^ab^	682 ± 198 ^b^	697 ± 92 ^b^	1016 ± 682 ^b^	752 ± 435 ^b^	601 ± 359 ^b^
*Senecio* *cineraria*	0 m	429 ± 176 ^a^	691 ± 235 ^a^	260 ± 153 ^a^	231 ± 159 ^a^	55 ± 26 ^a^	43 ± 50 ^a^	234 ± 138 ^a^	152 ± 48 ^a^
	3 m	2893 ± 3634 ^ab^	140 ± 54 ^b^	1686 ± 2174 ^ab^	344 ± 159 ^ab^	2450 ± 264 ^b^	2450 ± 343 ^c^	158 ± 83 ^a^	231 ± 114 ^a^
	6 m	2800 ± 884 ^b^	132 ± 44 ^b^	649 ± 242 ^b^	383 ± 55 ^ab^	1056 ± 538 ^b^	1047 ± 582 ^b^	623 ± 329 ^b^	548 ± 94 ^b^

Values are mean ± S.E. (n = 5). Values in a column followed by the same letter are not significantly different at *p* ≤ 0.05 in ANOVA and Tukey’s test.

**Table 2 plants-13-02519-t002:** Element concentrations (mg·kg^−1^) in leaf tissue of *Hedera helix* and *Senecio cineraria* by ICP.

Plant	Site	Time	Ca	Cd	Cu	Fe	K	Mg	Mn	Na	P	S	Zn
*Hedera helix*	Site 1	0 m	32,998.98 ± 27,408.30 ^a^	Nd	3.47 ± 0.62 ^a^	282.60 ± 84.91 ^a^	10,177.23 ± 2375.02 ^a^	2878.20 ± 632.54 ^a^	141.70 ± 106.07 ^a^	1960.00± 1227.32 ^a^	2860.35 ± 398.44 ^a^	2818.02± 613.94 ^a^	38.31 ± 20.02 ^a^
		3 m	24,550.20 ± 8769.40 ^a^	2.89 ± 0.28 ^a^	5.27 ± 2.37 ^ab^	278.83 ± 118.10 ^a^	13,758.05 ± 3142.70 ^a^	4427.88 ± 775.66 ^b^	170.41 ± 117.23 ^a^	1032.23 ± 342.76 ^a^	5956.04 ± 435.09 ^b^	3523.38 ± 767.15 ^a^	259.68 ± 73.00 ^b^
		6 m	25,406.80 ± 8920.33 ^a^	2.94 ± 0.27 ^a^	5.29 ± 2.37 ^ab^	283.83 ± 114.90 ^a^	13,846.05 ± 3147.12 ^a^	4445.88 ± 779.02 ^b^	171.21 ± 117.40 ^a^	1040.63 ± 343.05 ^a^	6002.24 ± 443.03 ^b^	3551.38 ± 793.25 ^a^	266.88 ± 71.05 ^b^
	Site 2	0 m	23,197.17 ± 7330.51 ^a^	Nd	4.81 ± 1.68 ^a^	383.47 ± 225.47 ^a^	9875.71 ± 1534.85	3350.94 ± 483.98 ^a^	249.40 ± 115.99 ^a^	1961.76 ± 895.02 ^a^	5049.78 ± 2329.52 ^b^	3380.00 ± 1388.36 ^a^	56.00 ± 22.86 ^a^
		3 m	48,032.26 ± 51,400.51 ^a^	4.23 ± 0.87 ^b^	7.32 ± 4.38 ^ab^	321.38 ± 70.78 ^a^	13,599.53 ± 4004.43 ^a^	3787.71 ± 1443.64 ^a^	90.95 ± 70.04 ^a^	902.90 ± 820.12 ^a^	5355.60 ± 1904.17 ^b^	2865.20 ± 973.93 ^a^	112.76 ± 90.17 ^a^
		6 m	19,401.31 ± 5444.44 ^a^	Nd	4.67 ± 1.29 ^a^	649.47 ± 317.21 ^ab^	11,185.56 ± 1650.25 ^a^	3288.92 ± 965.82 ^a^	75.17 ± 29.09 ^a^	797.30 ± 742.60 ^a^	3611.54 ± 719.13 ^a^	3833.01 ± 992.85 ^a^	31.64 ± 6.22 ^a^
	Site 3	0 m	25,929.29 ± 23,126.85 ^a^	Nd	3.21 ± 1.34 ^a^	221.79 ± 54.85 ^a^	11,111.98 ± 1172.28 ^a^	2355.56 ± 493.09 ^a^	46.03 ± 13.86 ^a^	3046.21 ± 2273.69 ^a^	3146.16 ± 636.60 ^a^	2010.47 ± 661.11 ^a^	43.11 ± 25.10 ^a^
		3 m	33,938.87 ± 38,909.77 ^a^	Nd	2.48 ± 0.80 ^a^	179.69 ± 52.53 ^a^	9353.99 ± 729.09 ^a^	2163.34 ± 570.93 ^a^	56.97 ± 12.56 ^a^	1236.40 ± 687.53 ^a^	1741.14 ± 400.99 ^a^	2453.05 ± 536.44 ^a^	49.58 ± 17.04 ^a^
		6 m	30,095.36 ± 22,843.27 ^a^	Nd	3.96 ± 2.98 ^a^	309.19 ± 95.20 ^a^	12,189.52 ± 3214.23 ^a^	3452.43 ± 1190.64 ^a^	349.01 ± 227.08 ^b^	1267.34 ± 1277.31 ^a^	5211.00 ± 1801.86 ^b^	4396.05 ± 2204.44 ^a^	85.82 ± 41.79 ^a^
	Site 4	0 m	43,931.55 ± 37,549.14 ^a^	Nd	2.20 ± 0.81 ^a^	247.16 ± 77.11 ^a^	13,281.14 ± 1698.19 ^a^	2130.03 ± 688.09 ^a^	101.30 ± 130.88 ^a^	2260.15 ± 1059.64 ^a^	4830.25 ± 2304.66 ^a^	1998.10 ± 1479.04 ^a^	41.17 ± 21.86 ^a^
		3 m	18,168.02 ± 6224.95 ^a^	Nd	2.25 ± 0.57 ^a^	215.74 ± 57.74 ^a^	13,351.28 ± 3530.92 ^a^	2526.43 ± 626.98 ^a^	107.40 ± 147.84 ^a^	1221.94 ± 602.59 ^a^	4110.69 ± 931.49 ^a^	2191.61 ± 1014.37 ^a^	63.94 ± 18.51 ^a^
		6 m	13,619.08 ± 5789.33 ^a^	0.32 ± 0.47 ^c^	6.03 ± 3.36 ^ab^	203.90 ± 57.85 ^a^	13,245.27 ± 5173.38 ^a^	2653.63 ± 507.95 ^a^	54.49 ± 60.57 ^a^	600.77 ± 560.78 ^a^	6978.94 ± 1613.66 ^a^	3121.97 ± 554.38 ^a^	72.78 ± 23.90 ^a^
*Senecio cineraria*	Site 1	0 m	43,003.87 ± 41,889.75 ^b^	Nd	3.47 ± 1.04 ^c^	235.83 ± 203.76 ^b^	13,576.72 ± 1680.00 ^b^	3377.58 ± 890.75 ^b^	91.99 ± 37.43 ^c^	7524.43 ± 4040.73 ^b^	3918.28 ± 1802.97 ^c^	3081.96 ± 739.87 ^b^	19.60 ± 2.89 ^c^
		3 m	87,656.37 ± 19,066.81 ^ab^	3.53 ± 0.54 ^d^	8.78 ± 2.91 ^d^	2190.97 ± 2106.59 ^d^	8507.05 ± 4559.51 ^b^	2050.04 ± 223.34 ^bc^	78.46 ± 65.85 ^c^	1896.24 ± 2402.75 ^c^	5738.16 ± 1423.92 ^c^	2119.96 ± 552.69 ^b^	1064.59 ± 712.99 ^d^
		6 m	88,316.37 ± 19,645.60 ^ab^	3.69 ± 0.63 ^d^	9.46 ± 3.17 ^d^	2216.97 ± 2112.24 ^d^	8565.05 ± 4585.26 ^b^	2098.04 ± 191.59 ^bc^	86.90 ± 72.92 ^c^	1941.84 ± 2435.27 ^c^	5822.16 ± 1434.75 ^c^	2185.96 ± 575.67 ^b^	1074.59 ± 712.99 ^d^
	Site 2	0 m	88,507.02 ± 16,117.84 ^ab^	Nd	4.24 ± 1.90 ^c^	730.86 ± 164.83 ^c^	10,782.14 ± 1315.85 ^b^	3018.09 ± 548.62 ^b^	73.32 ± 21.72 ^c^	8541.73 ± 2537.09 ^b^	3559.17 ± 310.35 ^c^	2963.39 ± 355.52 ^b^	17.86 ± 10.12 ^c^
		3 m	59,156.13 ± 38,898.75 ^b^	4.14 ± 0.64 ^d^	6.24 ± 2.60 ^cd^	2818.65 ± 2098.26 ^d^	16,404.67 ± 2925.56 ^c^	3944.36 ± 1815.69 ^b^	101.35 ± 65.43 ^c^	9709.67 ± 5393.01 ^b^	5175.05 ± 1605.05 ^c^	4113.68 ± 2126.64 ^b^	52.44 ± 29.48 ^c^
		6 m	18,171.57 ± 7147.07 ^b^	0.31 ± 0.20 ^e^	11.64 ± 3.28 ^d^	6084.31 ± 1533.21 ^d^	15,767.24 ± 3801.27 ^c^	6859.51 ± 1959.25 ^d^	162.11 ± 49.24 ^c^	14,061.89 ± 7121.0 ^b^	7615.10 ± 5031.09 ^c^	7365.76 ± 2770.72 ^c^	83.95 ± 29.79 ^d^
	Site 3	0 m	47,180.13 ± 33,223.00 ^b^	Nd	2.34 ± 1.05 ^c^	778.30 ± 188.21 ^c^	12,669.43 ± 2541.18 ^b^	3200.89 ± 901.71 ^b^	84.08 ± 33.99 ^c^	8664.28 ± 7280.34 ^b^	5181.14 ± 3379.99 ^c^	3079.68 ± 1357.29 ^b^	18.20 ± 4.22 ^c^
		3 m	39,425.96 ± 35,092.92 ^b^	Nd	4.36 ± 3.20 ^c^	1146.82 ± 394.87 ^d^	11,196.75 ± 3419.12 ^b^	3716.50 ± 1002.07 ^b^	81.51 ± 19.01 ^c^	7279.63 ± 1295.19 ^b^	3892.59 ± 827.82 ^c^	4391.16 ± 725.90 ^b^	29.80 ± 8.65 ^c^
		6 m	83,017.81 ± 78,166.32 ^ab^	Nd	6.98 ± 5.01 ^cd^	1109.90 ± 614.07 ^d^	14,541.05 ± 12,435.81 ^b^	4379.37 ± 2126.98 ^b^	109.61 ± 61.35 ^c^	7949.80 ± 3392.24 ^b^	8947.63 ± 6062.20 ^c^	6508.40 ± 3249.88 ^b^	61.93 ± 41.55 ^d^
	Site 4	0 m	37,841.02 ± 37,178.86 ^b^	Nd	4.23 ± 2.40 ^c^	324.66 ± 71.99 ^b^	14,974.45 ± 1196.15 ^b^	3917.13 ± 862.73 ^b^	126.87 ± 65.10 ^c^	12,689.41 ± 4629.01 ^b^	7830.11 ± 2236.98 ^c^	5344.27 ± 1095.24 ^b^	26.25 ± 7.07 ^c^
		3 m	50,865.89 ± 41,348.93 ^b^	Nd	4.76 ± 3.16 ^c^	282.47 ± 83.96 ^b^	14,674.16 ± 1360.47 ^b^	3796.20 ± 748.57 ^b^	54.37 ± 17.25 ^c^	9090.10 ± 2127.22 ^b^	5650.92 ± 1327.42 ^c^	5662.42 ± 1925.06 ^b^	53.36 ± 13.71 ^c^
		6 m	54,148.55 ± 39,324.89 ^b^	0.69 ± 0.13 ^e^	4.34 ± 2.33 ^c^	2084.22 ± 1236.25 ^d^	4875.76 ± 1035.63 ^d^	3567.87 ± 1182.56 ^b^	120.40 ± 67.42 ^c^	1360.68 ± 338.54 ^c^	4823.98 ± 1367.35 ^c^	2243.05 ± 529.26 ^b^	87.31 ± 45.13 ^d^

Values are mean (n = 5); values in a column followed by the same letter are not significantly different at *p* ≤ 0.05 in the Kruskal–Wallis and Tukey’s tests. Limits of detection (LD): Cd = 0.012 mg Kg^−1^, Cu = 0.006 mg Kg^−1^; Pb = 0.054 mg Kg^−1^; Zn = 0.16 mg Kg^−1^. nd: not detected.

## Data Availability

Data is contained within the article and Appendix A.

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
