# Peer review of "Quantification of Airborne Particulate Matter and Trace Element Deposition on Hedera helix and Senecio cineraria Leaves"

_plants, 2024, doi:10.3390/plants13172519_

Round 1

Reviewer 1 Report

Comments and Suggestions for Authors

This research is interesting. It focuses on environmental monitoring using potentially less expensive and more accessible matrices in the form of plant leaves. The paper is very well written. The study was carefully planned and conducted over a relatively long period. Numerous statistical analyses were performed, which is a significant strength of this research and contributes to my positive evaluation of the work. The literature is well selected, and the research techniques used are interesting. The discussion is engaging.

In this research, the element of novelty should certainly be emphasized more. The work has raised several questions, which I present below. My questions concern the general purpose of conducting this research. Addressing these questions and highlighting the significance of the research and plant selection is crucial for further evaluation of the paper. Additionally, the review includes recommendations for graphical changes in the manuscript.

Author Response

Comments 1: Why is Senecio cineraria, despite its specific shape, considered to be a good bioindicator? What do authors mean by “characteristics of its leaves make it a good candidate to sequester PM”?

Response 1: In line 70 is mentioned and cited that: Atmospheric dust deposition on leaves is mainly influenced by the plant species (ever-green or deciduous, composition and thickness of wax layer) and the specific structure of their leaves (e.g. leaf size, shape, roughness, presence of trichomes). Senecio cineraria has several of these characteristics (line 328). There’s no other reason despite its specific shape to choose Senecio for the experiment.

Comments 2: Regarding biomonitoring, the authors mention that it is limited to plants or lichens. What is the authors' opinion on biomonitoring using spider webs?

Response 2: Thank you for pointing this out. I have, accordingly modified the text (line 61).

Comments 3: Isn't such a strong influence of meteorological factors on the presence of particles on leaves completely disqualifying for the widespread use of these techniques? The authors mention the negative aspects of conventional monitoring, but a major advantage of it is that meteorological factors do not affect the accuracy of the obtained results. What do the authors think about this?

Response 3: I have added in Line 257, the arguments about why both technics are complementary.

Comments 4: Why is leaf sampling conducted only every 3 months? What is the shortest period (from literature) that the sampling is done?

Response 4: The shortest period in literature is one week, however we decided to conduce the experiment every 3 months duo to the time necessary to performed the analysis.

Comments 5: Fig. 6 is unreadable and somewhat sloppy; it needs to be improved. Some graphs have borders while others do not, and the triangles or circles are not visible. I would recommend presenting these results in a different format, such as a table or another clearer form. Alternatively, consider including enlarged versions of the graphs in the supplementary material.

Response 5: Agree. Figure 6 was chanced for Table 6.

Comments 6: The map with measurement stations/area of sampling is needed.

Response 6: Agree. The map with measurement station is now Figure 8.

Comments 7: The authors suggest that using standard monitoring generates high costs and energy consumption. However, they then assume that using biomonitoring, despite its limitations, could be used as a complementary tool. Doesn't it generate additional cost and energy consumption during analysis? Is it worthy?

Response 7: When we mentioned the energy consumption as a Drawbacks of automatic monitors, we were focusing on the use of them on field studies. We have clarified it in the text (line 60).

Comments 8: The conclusion states that the obtained concentration is species-specific, which suggests that this type of biomonitoring cannot be widely applied. There are likely many studies on this topic. Therefore, selecting another different leaves and conducting further studies might seem somewhat pointless. I would like the authors to prove me wrong.

Response 8: Certainly, despite every project may require the individual selection and calibration of different plant species, we value the adaptive and ecological advantages of using different native or local adapted species and see variability as the chance of tailor-made monitoring projects, “blending” monitor species with its surroundings when illegal emissions are tracked back. In that context, we think that although specie-specific response may increase labours in the experimental design and fine-tuning, it enhances the implementation chances. In additions, according to our experience, implementation cost and the opportunity of widespan of PM monitoring capacities make plants-based project to worth it.

We consider that is not about “prove the reviewer wrong” but conclude from a different perspective, interpreting the interspecies differences as biotechnological opportunities to choose from, favouring the use of native or locally adapted species. We understand the observation stated, regarding the limitations of species uniqueness on PM interaction and how it is perceived as a drawback to develop an “universal system" that could be directly set on the field, providing standard protocols and comparable data. Nevertheless, we observed that, even if most of plants performed in a similar fashion, environmental variables, local condition and intra specie variation also interfere in that conceptual model.

Reviewer 2 Report

Comments and Suggestions for Authors

Abstract: is good but too general, please use some numerical values to show the most important findings from your study

Introduction: you need to add another paragraph with some examples with higher plants and their leaves as a examples like bio-monitors; maybe you can also add some examples of comparing the bio-monitoring with classical monitoring like this (or in discussion chapter): https://doi.org/10.3390/ijerph19084706

Conclusions: too long, that section should be more critical with some numerical data from your research

Author Response

Comments 1: Abstract: is good but too general, please use some numerical values to show the most important findings from your study.

Response 1: Thank you for pointing this out. I have, accordingly modified the abstract.

Comments 2: Introduction: you need to add another paragraph with some examples with higher plants and their leaves as a examples like bio-monitors; maybe you can also add some examples of comparing the bio-monitoring with classical monitoring like this (or in discussion chapter): https://doi.org/10.3390/ijerph19084706

Response 2: A new paragraph was added to the introduction, addressing the reviewer's suggestions (line 74). Also, comparation between bio-monitoriong and classical monitoring was added in the discussion (line 261).

Comments 3: Conclusions: too long, that section should be more critical with some numerical data from your research.

Response 3: Agree. I have, accordingly modified the conclusions.